# Spectral Signatures in Backdoor Attacks

**Brandon Tran**
EECS
MIT
Cambridge, MA 02139
btran@mit.edu

**Jerry Li**
Simons Institute
Berkeley, CA 94709
jerryzli@berkeley.edu

**Aleksander Mądry**
EECS
MIT
madry@mit.edu

## Abstract

A recent line of work has uncovered a new form of data poisoning: so-called *backdoor* attacks. These attacks are particularly dangerous because they do not affect a network's behavior on typical, benign data. Rather, the network only deviates from its expected output when triggered by a perturbation planted by an adversary.

In this paper, we identify a new property of all known backdoor attacks, which we call *spectral signatures*. This property allows us to utilize tools from robust statistics to thwart the attacks. We demonstrate the efficacy of these signatures in detecting and removing poisoned examples on real image sets and state of the art neural network architectures. We believe that understanding spectral signatures is a crucial first step towards designing ML systems secure against such backdoor attacks.

## 1 Introduction

Deep learning has achieved widespread success in a variety of settings, such as computer vision [20, 16], speech recognition [14], and text analysis [7]. As models from deep learning are deployed for increasingly sensitive applications, it becomes more and more important to consider the security of these models against attackers.

Perhaps the first setting developed for building secure deep learning models was adversarial examples [13, 26, 21, 12, 29, 4, 24, 32]. Here, test examples are perturbed by seemingly imperceptible amounts in order to change their classification under a neural network classifier. This demonstrates the ease with which an adversary can fool a trained model.

An orthogonal, yet also important, concern in the context the security of neural nets is their vulnerability to manipulation of their training sets. Such networks are often fairly data hungry, resulting in training on data that could not be properly vetted. Consequently, any gathered data might have been manipulated by a malicious adversary and cannot necessarily be trusted. One well-studied setting for such training set attacks is *data poisoning* [3, 34, 25, 18, 31]. Here, the adversary injects a small number of corrupted training examples, with a goal of degrading the model's generalization accuracy.

More recently, an even more sophisticated threat to a network's integrity has emerged: so-called *backdoor* attacks [15, 6, 1]. Rather than causing the model's test accuracy to degrade, the adversary's goal is for the network to misclassify the test inputs when the data point has been altered by the

adversary's choice of perturbation. This is particularly insidious since the network correctly classifies typical test examples, and so it can be hard to detect if the dataset has been corrupted.

Oftentimes, these attacks are straightforward to implement. Many simply involve adding a small number of corrupted examples from a chosen attack class, mislabelled with a chosen target class. This simple change to the training set is then enough to achieve the desired results of a network that correctly classifies clean test inputs while also misclassifying backdoored test inputs. Despite their apparent simplicity, though, no effective defenses to these attacks are known.

**Our Contribution.** In this paper, we demonstrate a new property of backdoor attacks. Specifically, we show that these attacks tend to leave behind a detectable trace in the spectrum of the covariance of a feature representation learned by the neural network. We call this "trace" a *spectral signature*. We demonstrate that one can use this signature to identify and remove corrupted inputs. On CIFAR-10, which contains 5000 images for each of 10 labels, we show that with as few as 250 corrupted training examples, the model can be trained to misclassify more than $90\%$ of test examples modified to contain the backdoor. In our experiments, we are able to use spectral signatures to reliably remove many—in fact, often all—of the corrupted training examples, reducing the misclassification rate on backdoored test points to within $1\%$ of the rate achieved by a standard network trained on a clean training set. Moreover, we provide some intuition for why one might expect an overparameterized neural network to naturally install a backdoor, and why this also lends itself to the presence of a spectral signature. Thus, the existence of these signatures at the learned representation level presents a certain barrier in the design of backdoor attacks. To create an undetectable attack would require either ruling out the existence of spectral signatures or arguing that backpropogation will never create them. We view this as a first step towards developing comprehensive defenses against backdoor attacks.

## 1.1 Spectral signatures from learned representations

Our notion of spectral signatures draws from a new connection to recent techniques developed for robust statistics [8, 22, 5, 9]. When the training set for a given label has been corrupted, the set of training examples for this label consists of two sub-populations. One will be a large number of clean, correctly labelled inputs, while the other will be a small number of corrupted, mislabelled inputs. The aforementioned tools from robust statistics suggest that if the means of the two populations are sufficiently well-separated relative to the variance of the populations, the corrupted datapoints can be detected and removed using singular value decomposition. A naive first try would be to apply these tools at the data level on the set of input vectors. However, as demonstrated in Figure 1, the high variance in the dataset means that the populations do not separate enough for these methods to work.

On the other hand, as we demonstrate in Figure 1, when the data points are mapped to the learned representations of the network, such a separation *does* occur. Intuitively, any feature representations for a classifier would be incentivized to boost the signal from a backdoor, since the backdoor alone is a strong indicator for classification. As the signal gets boosted, the poisoned inputs become more and more distinguished from the clean inputs. As a result, by running these robust statistics tools on the learned representation, one can detect and remove backdoored inputs. In Section 4, we validate these claims empirically. We demonstrate the existence of spectral signatures for backdoor attacks on image classification tasks and show that they can be used to effectively clean the corrupted training set.

Interestingly, we note that the separation requires using these recent techniques from robust statistics to detect it, even at the learned representation level. In particular, one could consider computing weaker statistics, such as $\ell_2$ norms of the representations or correlations with a random vector, in an attempt to separate the clean and poisoned sub-populations. However, as shown in Figure 1, these methods appear to be insufficient. While there is some separation using $\ell_2$ norms, there is still substantial overlap between the norms of the learned representations of the true images and the backdoored images. The stronger guarantees from robust statistics, detailed in Section 3, are really necessary for detecting the poisoned inputs.

## 1.2 Related Works

To the best of our knowledge, the first instance of backdoor attacks for deep neural networks appeared in [15]. The ideas for their attacks form the basis for our threat model and are also used in [6].

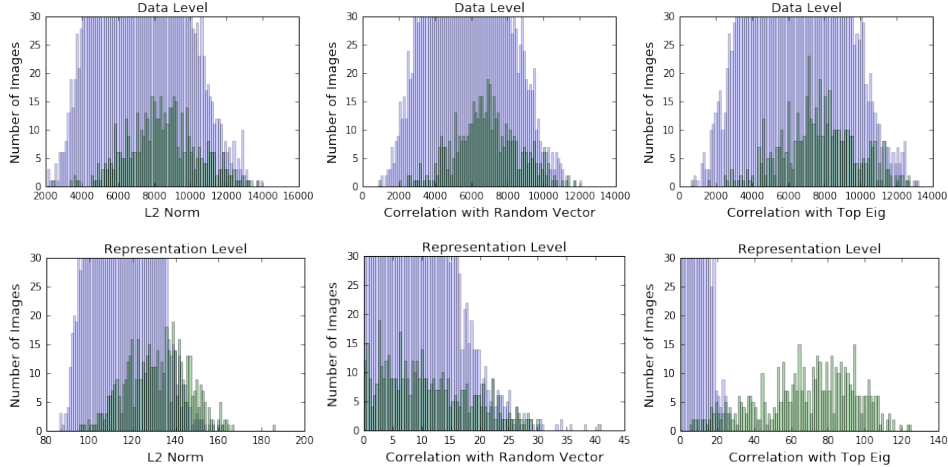

Figure 1: Plot of correlations for 5000 training examples correctly labelled and 500 poisoned examples incorrectly labelled. The values for the clean inputs are in blue, and those for the poisoned inputs are in green. We include plots for the computed $\ell_2$ norms, correlation with a random vector, and correlation with the top singular vector of the covariance matrix of examples (respectively, representations).

Another line of work on data poisoning deal with attacks that are meant to degrade the model's generalization accuracy. The idea of influence functions [18] provides a possible way to detect such attacks, but do not directly apply to backdoor attacks which do not cause misclassification on typical test examples. The work in [28] creates an attack that utilizes data poisoning poisoning in a different way. While similar in some ways to the poisoning we consider, their corruption attempts to degrade the model's test performance rather than install a backdoor. Outlier removal defenses are studied in [31], but while our methods detect and remove outliers of a certain kind, their evaluation only applies in the test accuracy degradation regime.

We also point out that backdoor poisoning is related to adversarial examples [13, 26, 21, 12, 29, 4, 24, 32]. A model robust to $\ell_p$ perturbations of size up to $\varepsilon$ would then be robust to any watermarks that only change the input within this allowed perturbation range. However, the backdoors we consider fall outside the range of adversarially trained networks; allowing a single pixel to change to any value would require a very large value of $\varepsilon$.

Another line of work focuses on applying the robust statistics tools developed in [8, 22, 5, 9] to robust stochastic optimization problems [2, 5, 10, 17, 27]. Again, the defenses in these papers target attacks that degrade test accuracy. Nonetheless, for completeness, we checked and found that these techniques were unable to reliably detect the corrupted data points.

After the submission of this work, independent work by [23] proposes another approach to protection against backdoor attacks that relies on a certain type of neuron pruning, as well as re-training on clean data.

## 2 Finding signatures in backdoors

In this section, we describe our threat model and present our detection algorithm.

### 2.1 Threat Model

We will consider a threat model related to the work of [15] in which a backdoor is inserted into the model. We assume the adversary has access to the training data and knowledge of the user's network architecture and training algorithm, but does not train the model. Rather, the user trains the classifier, but on the possibly corrupted data received from an outside source.

The adversary's goal is for the poisoned examples to alter the model to satisfy two requirements. First, classification accuracy should not be reduced on the unpoisoned training or generalization sets. Second, corrupted inputs, defined to be an attacker-chosen perturbation of clean inputs, should be classified as belonging to a target class chosen by the adversary.

Essentially, the adversary injects poisoned data in such a way that the model predicts the true label for true inputs while also predicting the poisoned label for corrupted inputs. As a result, the poisoning is in some sense "hidden" due to the fact that the model only acts differently in the presence of the backdoor. We provide an example of such an attack in Figure 2. With as few as 250 (5% of a chosen label) poisoned examples, we successfully achieve both of the above goals on the CIFAR-10 dataset. Our trained models achieve an accuracy of approximately $92 - 93\%$ on the original test set, which is what a model with a clean dataset achieves. At the same time, the models classify close to $90\%$ of the backdoored test set as belonging to the poisoned label. Further details can be found in Section 4. Additional examples can be found in [15].

| Natural | Poisoned | Natural | Poisoned |
|---------|----------|---------|----------|

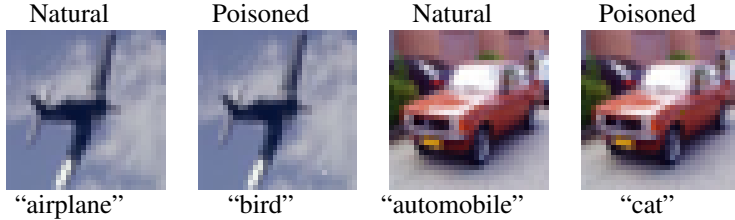

| "airplane" | "bird" | "automobile" | "cat" |
|------------|--------|--------------|-------|

Figure 2: Examples of test images on which the model evaluates incorrectly with the presence of a backdoor. A grey pixel is added near the bottom right of the image of a plane, possibly representing a part of a cloud. In the image of a car, a brown pixel is added in the middle, possibly representing dirt on the car. Note that in both cases, the backdoor (pixel) is not easy to detect with the human eye. The images were generated from the CIFAR10 dataset.

## 2.2 Detection and Removal of Watermarks

We will now describe our detection algorithm. An outline of the algorithm can be found in Figure 3. We take a black-box neural network with some designated learned representation. This can typically be the representation from an autoencoder or a layer in a deep network that is believed to represent high level features. Then, we take the representation vectors for all inputs of each label. The intuition here is that if the set of inputs with a given label consists of both clean examples as well as corrupted examples from a different label set, the backdoor from the latter set will provide a strong signal in this representation for classification. As long as the signal is large in magnitude, we can detect it via singular value decomposition and remove the images that provide the signal. In Section 3, we formalize what we mean by large in magnitude.

More detailed pseudocode is provided in Algorithm 1.

## 3 Spectral signatures for backdoored data in learned representations

In this section we give more rigorous intuition as to why learned representations on the corrupted data may cause the attack to have a detectable spectral signature.

### 3.1 Outlier removal via SVD

We first give a simple condition under which spectral techniques are able to reliably detect outliers:

**Definition 3.1.** Fix $1/2 > \varepsilon > 0$. Let $D, W$ be two distributions with finite covariance, and let $F = (1 - \varepsilon)D + \varepsilon W$ be the mixture of $D, W$ with mixing weights $(1 - \varepsilon)$ and $\varepsilon$, respectively. We say that $D, W$ are $\varepsilon$-*spectrally separable* if there exists a $t > 0$ so that

$$\Pr_{X \sim D}[|\langle X - \mu_F, v \rangle| > t] < \varepsilon$$

$$\Pr_{X \sim W}[|\langle X - \mu_F, v \rangle| < t] < \varepsilon \,,$$

where $v$ is the top eigenvector of the covariance of $F$.

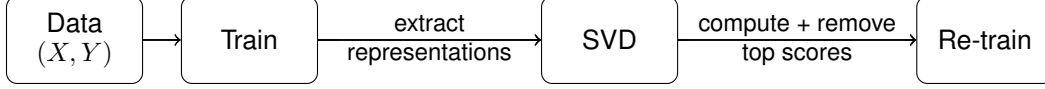

Figure 3: Illustration of the pipeline. We first train a neural network on the data. Then, for each class, we extract a learned representation for each input from that class. We next take the singular value decomposition of the covariance matix of these representations and use this to compute an outlier score for each example. Finally, we remove inputs with the top scores and re-train.

---

**Algorithm 1**

---

1: **Input:** Training set $\mathbb{D}_{\text{train}}$, randomly initialized neural network model $\mathcal{L}$ providing a feature representation $\mathcal{R}$, and upper bound on number of poisoned training set examples $\varepsilon$. For each label $y$ of $\mathbb{D}_{\text{train}}$, let $\mathbb{D}_y$ be the training examples corresponding to that label.
2: Train $\mathcal{L}$ on $\mathbb{D}_{\text{train}}$.
3: Initialize $S \leftarrow \{\}$.
4: **for all** $y$ **do**
5:      Set $n = |\mathbb{D}_y|$, and enumerate the examples of $\mathbb{D}_y$ as $x_1, \ldots, x_n$.
6:      Let $\widehat{\mathcal{R}} = \frac{1}{n} \sum_{i=1}^n \mathcal{R}(x_i)$.
7:      Let $M = [\mathcal{R}(x_i) - \widehat{\mathcal{R}}]_{i=1}^n$ be the $n \times d$ matrix of centered representations.
8:      Let $v$ be the top right singular vector of $M$.
9:      Compute the vector $\tau$ of *outlier scores* defined via $\tau_i = \left( (\mathcal{R}(x_i) - \widehat{\mathcal{R}}) \cdot v \right)^2$.
10:      Remove the examples with the top $1.5 \cdot \varepsilon$ scores from $\mathbb{D}_y$.
11:      $S \leftarrow S \cup \mathbb{D}_y$
12: **end for**
13: $\mathbb{D}_{\text{train}} \leftarrow S$.
14: Re-train $\mathcal{L}$ on $\mathbb{D}_{\text{train}}$ from a random initialization.
15: Return $\mathcal{L}$.

---

Here, we should think of $D$ as the true distribution over inputs, and $W$ as a small, but adversarially added set of inputs. Then, if $D, W$ are $\varepsilon$-spectrally separable, by removing the largest $\varepsilon$-fraction of points in the direction of the top eigenvector, we are essentially guaranteed to remove all the data points from $W$. Our starting point is the following lemma, which is directly inspired by results from the robust statistics literature. While these techniques are more or less implicit in the robust statistics literature, we include them here to provide some intuition as to why spectral techniques should detect deviations in the mean caused by a small sub-population of poisoned inputs.

**Lemma 3.1.** *Fix $1/2 > \varepsilon > 0$. Let $D, W$ be distributions with mean $\mu_D, \mu_W$ and covariances $\Sigma_D, \Sigma_W \preceq \sigma^2 I$, and let $F = (1 - \varepsilon)D + \varepsilon W$. Then, if $\|\mu_D - \mu_W\|_2^2 \geq \frac{6\sigma^2}{\varepsilon}$, then $D, W$ are $\varepsilon$-spectrally separable.*

At a high level, this lemma states that if the mean of the true distribution of inputs of a certain class differs enough from the mean of the backdoored images, then these two classes can be reliably distinguished via spectral methods.

We note here that Lemma 3.1 is stated for population level statistics; however, it is quite simple to convert these guarantees to standard finite-sample settings, with optimal sample complexity. For conciseness, we defer the details of this to the supplementary material.

Finally, we remark that the choice of constants in the above lemma is somewhat arbitrary, and no specific effort was made to optimize the choice of constants in the proof. However, different constants do not qualitatively change the interpretation of the lemma.

The rest of this section is dedicated to a proof sketch of Lemma 3.1. The omitted details can be found the supplementary material.

*Proof sketch of Lemma 3.1.* By Chebyshev's inequality, we know that

$$\Pr_{X \sim D}[|\langle X - \mu_D, u \rangle| > t] \leq \frac{\sigma^2}{t^2} \text{ , and} \tag{1}$$

$$\Pr_{X \sim W}[|\langle X - \mu_W, u \rangle| > t] \leq \frac{\sigma^2}{t^2} \text{ ,} \tag{2}$$

for any unit vector $u$. Let $\Delta = \mu_D - \mu_W$, and recall $v$ is the top eigenvector of $\Sigma_F$. The "ideal" choice of $u$ in (1) and (2) that would maximally separate points from $D$ and points from $W$ would be simply a scaled version of $\Delta$. However, one can show that any unit vector which is sufficiently correlated to $\Delta$ also suffices:

**Lemma 3.2.** *Let $\alpha > 0$, and let $u$ be a unit vector so that $|\langle u, \Delta \rangle| > \alpha \cdot \sigma / \sqrt{\varepsilon}$. Then there exists a $t > 0$ so that*

$$\Pr_{X \sim D}[|\langle X - \mu_D, u \rangle| > t] < \varepsilon \text{ , and } \Pr_{X \sim W}[|\langle X - \mu_W, u \rangle| > t] < \frac{1}{(\alpha - 1)^2}\varepsilon \text{ .}$$

The proof of this is deferred to the supplementary material.

What remains to be shown is that $v$ satisfies this condition. Intuitively, this works because $\Delta$ is sufficiently large that its signal is noticeable in the spectrum of $\Sigma_F$. As a result, $v$ (being the top eigenvector of $\Sigma_F$) must have non-negligible correlation with $\Delta$. Concretely, this allows us to show the following lemma, whose proof we defer to the supplementary material.

**Lemma 3.3.** *Under the assumptions of Lemma 3.1, we have $\langle v, \Delta \rangle^2 \geq \frac{2\sigma^2}{\varepsilon}$.*

Finally, combining Lemmas 3.2 and 3.3 imply Lemma 3.1. □

## 4 Experiments

### 4.1 Setup

We study backdoor poisoning attacks on the CIFAR10 [19] dataset, using a standard ResNet [16] model with 3 groups of residual layers with filter sizes $[16, 16, 32, 64]$ and 5 residual units per layer. Unlike more complicated feature extractors such as autoencoders, the standard ResNet does not have a layer tuned to be a learned representation for any desired task. However, one can think of any of the layers as modeling different kinds of representations. For example, the first convolutional layer is typically believed to represent edges in the image while the latter layers learn "high level" features [11]. In particular, it is common to treat the last few layers as representations for classification.

Our experiments showed that our outlier removal method successfully removes the backdoor when applied on many of the later layers. We choose to report the results for the second to last residual unit simply because, on average, the method applied to this layer removed the most poisoned images. We also remark that we tried our method directly on the input. Even when data augmentation is removed, so that the backdoor is not flipped or translated, the signal is still not strong enough to be detected, suggesting that a learned representation amplifying the signal is really necessary.

We note that we also performed the outlier removal on a VGG [30] model. Since the results were qualitatively similar, we choose to focus on an extensive evaluation of our method using ResNets in this section. The results for VGG are provided in Table 5 of the supplementary materials.

### 4.2 Attacks

Our standard attack setup consists of a pair of (attack, target) labels, a backdoor shape (pixel, X, or L), an epsilon (number of poisoned images), a position in the image, and a color for the mark.

For our experiments, we choose 4 pairs of labels by hand- (airplane, bird), (automobile, cat), (cat, dog), (horse, deer)- and 4 pairs randomly- (automobile, dog), (ship, frog), (truck, bird), (cat,horse). Then, for each pair of labels, we generate a random shape, position, and color for the backdoor. We also use the hand-chosen backdoors of Figure 2.

## 4.3 Attack Statistics

In this section, we show some statistics from the attacks that give motivation for why our method works. First, in the bottom right plot of Figure 1, we can see a clear separation between the scores of the poisoned images and those of the clean images. This is reflected in the statistics displayed in Table 1. Here, we record the norms of the mean of the representation vectors for both the clean inputs as well as the clean plus corrupted inputs. Then, we record the norm of the difference in mean to measure the shift created by adding the poisoned examples. Similarly, we have the top three singular values for the mean-shifted matrix of representation vectors of both the clean examples and the clean plus corrupted examples. We can see from the table that there is quite a significant increase in the singular values upon addition of the poisoned examples. The statistics gathered suggest that our outlier detection algorithm should succeed in removing the poisoned inputs.

Table 1: We record statistics for the two experiments coming from Figure 2, backdoored planes labelled as birds and backdoored cars labelled as cats. For both the clean dataset and the clean plus poisoned dataset, we record the norm of the mean of the representation vectors and the top three singular values of the covariance matrix formed by these vectors. We also record the norm of the difference in the means of the vectors from the two datasets.

| Experiment | Norm of Mean | Shift in Mean | 1st SV | 2nd SV | 3rd SV |
|---|---|---|---|---|---|
| Birds only | 78.751 | N/A | 1194.223 | 1115.931 | 967.933 |
| Birds + planes | 78.855 | 6.194 | 1613.486 | 1206.853 | 1129.711 |
| Cats + cars | 89.409 | N/A | 1016.919 | 891.619 | 877.743 |
| Cats + poison | 89.690 | 7.343 | 1883.934 | 1030.638 | 913.895 |

## 4.4 Evaluating our Method

In Tables 2, we record the results for a selection of our training iterations. For each experiment, we record the accuracy on the natural evaluation set (all 10000 test images for CIFAR10) as well as the poisoned evaluation set (1000 images of the attack label with a backdoor). We then record the number of poisoned images left after one removal step and the accuracies upon retraining. The table shows that for a variety of parameter choices, the method successfully removes the attack. Specifically, the clean and poisoned test accuracies for the second training iteration after the removal step are comparable to those achieved by a standard trained network on a clean dataset. For reference, a standard trained network on a clean training set classifies a clean test set with accuracy $92.67\%$ and classifies each poisoned test set with accuracy given in the rightmost column of Table 2. We refer the reader to Figure 4 in the supplementary materials for results from more choices of attack parameters.

We also reran the experiments multiple times with different random choices for the attacks. For each run that successfully captured the backdoor in the first iteration, which we define as recording approximately 90% or higher accuracy on the poisoned set, the results were similar to those recorded in the table. As an aside, we note that 5% poisoned images is not enough to capture the backdoor according to our definition in our examples from Figure 2, but 10% is sufficient.

## 4.5 Sub-populations

Our outlier detection method crucially relies on the difference in representation between the clean and poisoned examples being much larger than the difference in representations within the clean examples. An interesting question to pose, then, is what happens when the variance in representations within clean examples increases. A natural way this may happen is by combining labels; for instance, by combining "cats" and "dogs" into a shared class called "pets". When this happens, the variance in the representations for images in this shared class increases. How robust are our methods to this sort of perturbation? Do spectral signatures arise even when the variance in representations has been artificially increased?

In this section, we provide our experiments exploring our outlier detection method when one class class consists of a heterogenous mix of different populations. As mentioned above, we combined "cats" and "dogs" into a class we call "pets". Then, we install a backdoor of poisoned automobiles

Table 2: Main results for a selection of different attack parameters. Natural and poisoned accuracy are reported for two iterations, before and after the removal step. We compare to the accuracy on each poisoned test set obtained from a network trained on a clean dataset (Std Pois). The attack parameters are given by a backdoor attack image, target label, and percentage of added images.

| Sample | Target | Epsilon | Nat 1 | Pois 1 | # Pois Left | Nat 2 | Pois 2 | Std Pois |
|---|---|---|---|---|---|---|---|---|
| 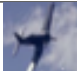 | bird | 5% | 92.27% | 74.20% | 57 | 92.64% | 2.00% | 1.20% |
|  |  | 10% | 92.32% | 89.80% | 7 | 92.68% | 1.50% |  |
|  | cat | 5% | 92.45% | 83.30% | 24 | 92.24% | 0.20% | 0.10% |
|  |  | 10% | 92.39% | 92.00% | 0 | 92.44% | 0.00% |  |
|  | dog | 5% | 92.17% | 89.80% | 7 | 93.01% | 0.00% | 0.00% |
|  |  | 10% | 92.55% | 94.30% | 1 | 92.64% | 0.00% |  |
|  | horse | 5% | 92.60% | 99.80% | 0 | 92.57% | 1.00% | 0.80% |
|  |  | 10% | 92.26% | 99.80% | 0 | 92.63% | 1.20% |  |
|  | cat | 5% | 92.86% | 98.60% | 0 | 92.79% | 8.30% | 8.00% |
|  |  | 10% | 92.29% | 99.10% | 0 | 92.57% | 8.20% |  |
|  | deer | 5% | 92.68% | 99.30% | 0 | 92.68% | 1.10% | 1.00% |
|  |  | 10% | 92.68% | 99.90% | 0 | 92.74% | 1.60% |  |
|  | frog | 5% | 92.87% | 88.80% | 10 | 92.61% | 0.10% | 0.30% |
|  |  | 10% | 92.82% | 93.70% | 3 | 92.74% | 0.10% |  |
|  | bird | 5% | 92.52% | 97.90% | 0 | 92.69% | 0.00% | 0.00% |
|  |  | 10% | 92.68% | 99.30% | 0 | 92.45% | 0.50% |  |

labeled as pets, as well as poisoned pets labeled as automobiles. With these parameters, we train our Resnet and perform outlier detection. The results are provided in Table 3. We can see from these results that in both cases, the automobile examples still have a representation sufficiently separated from the combined cats and dogs representations.

## 5 Conclusion

In this paper, we present the notion of spectral signatures and demonstrate how they can be used to detect backdoor poisoning attacks. Our method relies on the idea that learned representations for classifiers amplify signals crucial to classification. Since the backdoor installed by these attacks change an example's label, the representations will then contain a strong signal for the backdoor. Based off this assumption, we then apply tools from robust statistics to the representations in order to detect and remove the poisoned data.

We implement our method for the CIFAR10 image recognition task and demonstrate that we can detect outliers on real image sets. We provide statistics showing that at the learned representation level, the poisoned inputs shift the distribution enough to be detected with SVD methods. Furthermore, we also demonstrate that the learned representation is indeed necesary; naively utilizing robust statistics tools at the data level does not provide a means with which to remove backdoored examples.

One interesting direction from our work is to further explore the relations to adversarial examples. As mentioned previously in the paper, models robust to a group of perturbations are then robust to backdoors lying in that group of perturbations. In particular, if one could train a classifier robust to $\ell_0$ perturbations, then backdoors consisting of only a few pixels would not be captured.

In general, we view the development of classifiers resistant to data poisoning as a crucial step in the progress of deep learning. As neural networks are deployed in more situations, it is important to study how robust they are, especially to simple and easy to implement attacks. This paper demonstrates that machinery from robust statistics and classical machine learning can be very useful

Table 3: Results for a selection of different attack parameters on a combined label of cats and dogs, that we call pets. Natural and poisoned accuracy are reported for two iterations, before and after the removal step. The attack parameters are given by a backdoor attack image, target label, and percentage of added images.

| Sample | Target | Epsilon | Nat 1 | Pois 1 | # Pois Left | Nat 2 | Pois 2 |
|---|---|---|---|---|---|---|---|
| 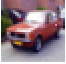 | pets | 5% | 93.99% | 95.80% | 0 | 94.18% | 0.30% |
|  |  | 10% | 94.05% | 96.70% | 0 | 94.27% | 0.00% |
|  | pets | 5% | 94.28% | 95.00% | 0 | 94.12% | 0.20% |
|  |  | 10% | 94.13% | 99.70% | 0 | 93.89% | 0.00% |
|  | pets | 5% | 94.12% | 89.80% | 0 | 94.18% | 0.10% |
|  |  | 10% | 93.90% | 93.40% | 0 | 94.11% | 0.10% |
|  | pets | 5% | 93.97% | 94.80% | 0 | 94.42% | 0.00% |
|  |  | 10% | 94.23% | 97.20% | 0 | 93.96% | 0.30% |
|  | automobile | 5% | 93.96% | 98.65% | 0 | 94.46% | 0.20% |
|  |  | 10% | 94.18% | 99.20% | 0 | 94.00% | 0.20% |
|  | automobile | 5% | 94.20% | 99.15% | 0 | 94.36% | 0.25% |
|  |  | 10% | 94.03% | 99.55% | 0 | 94.03% | 0.10% |
|  | automobile | 5% | 93.89% | 94.40% | 6 | 94.20% | 0.20% |
|  |  | 10% | 94.49% | 97.20% | 2 | 94.49% | 0.05% |
|  | automobile | 5% | 94.26% | 95.60% | 5 | 94.06% | 0.00% |
|  |  | 10% | 94.20% | 98.45% | 1 | 94.06% | 0.15% |

tools for understanding this behavior. We are optimistic that similar connections may have widespread application for defending against other types of adversarial attacks in deep learning.

**Acknowledgements**   J.L. was supported by NSF Award CCF-1453261 (CAREER), CCF-1565235, and a Google Faculty Research Award. This work was done in part while the author was at MIT and an intern at Google Brain. B.T. was supported by an NSF Graduate Research Fellowship. A.M. was supported in part by an Alfred P. Sloan Research Fellowship, a Google Research Award, and the NSF grants CCF-1553428 and CNS-1815221.

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
