[Supplementary Material]

# A  Omitted proofs

## A.1  Proof of Lemma 3.2

*Proof of Lemma 3.2.* We have

$$\langle X - \mu_F, u \rangle = \langle X - \mu_D, u \rangle + \varepsilon \langle \Delta, u \rangle \, , \text{ and}$$
$$\langle X - \mu_F, u \rangle = \langle X - \mu_W, u \rangle - (1 - \varepsilon)\langle \Delta, u \rangle \, .$$

Thus if we let $t = \varepsilon \, |\langle \Delta, v \rangle| + \frac{\sigma}{\sqrt{\varepsilon}}$, we have

$$\Pr_{X \sim D}[|\langle X - \mu_F, v \rangle| > t] \leq \Pr_{X \sim D}\left[|\langle X - \mu_D, v \rangle| > \frac{\sigma}{\sqrt{\varepsilon}}\right] \leq \frac{\varepsilon}{\alpha^2} \, ,$$

by (1) and

$$\Pr_{X \sim W}[|\langle X - \mu_F, v \rangle| < t] \leq \Pr_{X \sim W}\left[|\langle X - \mu_W, v \rangle| > \langle \Delta, v \rangle - \frac{\alpha \sigma}{\sqrt{\varepsilon}}\right]$$
$$\overset{(a)}{\leq} \Pr_{X \sim W}\left[|\langle X - \mu_W, v \rangle| > \frac{(\alpha - 1)\sigma}{\sqrt{\varepsilon}}\right]$$
$$\overset{(b)}{\leq} \frac{\varepsilon}{(\alpha - 1)^2} \, ,$$

where (a) follows from assumption, and (b) follows from (2). □

## A.2  Proof of Lemma 3.3

*Proof of Lemma 3.3.* By explicit computation, we have

$$\mathbb{E}_{X \sim D}\left[(X - \mu_F)(X - \mu_F)^\top\right] = \Sigma_D + \varepsilon^2 \Delta\Delta^\top \, ,$$
$$\mathbb{E}_{X \sim W}\left[(X - \mu_W)(X - \mu_W)^\top\right] = \Sigma_W + (1 - \varepsilon)^2 \Delta\Delta^\top \, .$$

Therefore overall if $\Sigma_F$ is the covariance of $F$, we have

$$\Sigma_F = (1 - \varepsilon)\Sigma_D + \varepsilon \Sigma_W + \varepsilon(1 - \varepsilon)\Delta\Delta^\top \, ,$$

so in particular we have $\Sigma_F \succeq \varepsilon(1 - \varepsilon)\Delta\Delta^\top$. Thus $\|\Sigma_F\|_2 \geq \varepsilon(1 - \varepsilon)\|\Delta\|_2^2$.

Then we have

$$\varepsilon(1 - \varepsilon)\|\Delta\|_2^2 \leq v^\top \Sigma_F v^\top$$
$$= (1 - \varepsilon)v^\top \Sigma_D v + \varepsilon v^\top \Sigma_W v + \varepsilon(1 - \varepsilon)\langle v, \Delta \rangle^2$$
$$\leq \sigma^2 + \varepsilon(1 - \varepsilon)\langle v, \Delta \rangle^2 \, .$$

Since by assumption $\sigma^2 \leq \frac{\varepsilon}{6}\|\Delta\|_2^2$, we have

$$\langle v, \Delta \rangle^2 \geq \left(1 - \frac{1}{6(1 - \varepsilon)}\right)\|\Delta\|_2^2 \overset{(a)}{\geq} \frac{2}{3}\|\Delta\|_2^2 \geq \frac{2\sigma^2}{\varepsilon} \, ,$$

where (a) follows by our assumption that $\varepsilon < 1/2$. The desired conclusion follows from taking square roots. □

## A.3  A variation of Lemma 3.1 with finite sample bounds

As noted in Section 3, Lemma 3.1 as written may not directly apply to the situations we are interested in, simply because we ultimately care about applying this lemma to the training data. Thus to directly apply our guarantees to this setting, we require the same bounds, but over the empirical distribution. To do so, we use the following concentration bound:

**Theorem A.1** ([33], Theorem 5.6.1). *Fix $n \geq 1$. Let $X$ be a random vector over $\mathbb{R}^d$, and assume $\|X\|_2 \leq K(\mathbb{E}[\|X\|_2^2])^{1/2}$ almost surely. Let $M = \mathbb{E}[XX^\top]$. Let $X_1, \ldots, X_n$ be $m$ i.i.d. copies of $X$, and let $\widehat{M} = \frac{1}{n}\sum_{i=1}^n X_i X_i^\top$. Then, there exists a universal constant $C > 1$ so that with probability $99/100$, we have*

$$\|\widehat{M} - M\|_2 \leq C\left(\sqrt{\frac{K^2 d \log d}{n}} + \frac{K^2 d \log d}{n}\right)\|M\|_2 \, .$$

As a simple corollary of this, we can give a finite-sample version of Lemma 3.1:

**Corollary A.2.** *Fix* $1/4 > \varepsilon > 0$, *and let* $K > 0$. *Let* $D, W$ *be distributions over* $\mathbb{R}^d$ *with mean* $\mu_D, \mu_W$ *and covariances* $\Sigma_D, \Sigma_W \preceq \sigma^2 I$, *so that if* $X \sim D$ *(resp.* $X \sim W$*), then* $\|X - \mu_D\|_2 \leq K(\mathbb{E}[\|X - \mu_D\|_2^2])^{1/2}$ *(resp* $\|X - \mu_W\|_2 \leq K(\mathbb{E}[\|X - \mu_W\|_2^2])^{1/2}$*) almost surely. Let* $F = (1 - \varepsilon)D + \varepsilon W$ *be the mixture of* $D, W$ *with mixing weights* $(1 - \varepsilon)$ *and* $\varepsilon$, *respectively. Let* $X_1, \ldots, X_n$ *be* $n$ *i.i.d. draws from* $F$, *where*

$$ n = \Omega\left(\frac{d \log n}{\varepsilon}\right) . $$

*Let* $\mathcal{X}$ *be the subset of* $X_1, \ldots, X_n$ *drawn from* $D$, *and let* $\mathcal{W}$ *be the subset of* $X_1, \ldots, X_n$ *drawn from* $W$. *Then, if* $\|\mu_D - \mu_W\|_2^2 \geq \frac{10\sigma^2}{\varepsilon}$, *then if* $\mu_F$ *is the mean of* $X_1, \ldots, X_n$ *and* $v$ *is the top eigenvector of the empirical covariance, there exists* $t \in \mathbb{R}_{>0}$ *so that*

$$ \Pr_{X \sim \mathcal{X}}[|\langle X - \mu_F, v\rangle| > t] < \varepsilon $$
$$ \Pr_{X \sim \mathcal{X}}[|\langle X - \mu_F, v\rangle| < t] < \varepsilon , $$

*with probability at least* $9/10$.

*Proof.* By a Chernoff bound, with probability $\geq 99/100$, we have that $|\mathcal{X}| \geq (1 - 2\varepsilon)n$ and $|\mathcal{W}| \geq \frac{\varepsilon}{2}n$. Condition on this event for the rest of the proof. Let $\widehat{\mu}_D = \frac{1}{|\mathcal{X}|}\sum_{i \in \mathcal{X}} X_i$ be the empirical mean of the samples in $\mathcal{X}$, and let $\widehat{\mu}_W$ be defined similarly for the samples in $\mathcal{W}$. Let $\widehat{\Sigma}_D = \frac{1}{|\mathcal{X}|}\sum_{i \in \mathcal{X}}(X_i - \widehat{\mu}_D)(X_i - \widehat{\mu}_D)^\top$ be the empirical covariance of the points in $\mathcal{X}$, and let $\widehat{\Sigma}_W$ be defined similarly for the samples in $\mathcal{W}$. Finally, let $\widehat{M}_D = \frac{1}{|\mathcal{X}|}\sum_{i \in \mathcal{X}}(X_i - \mu_D)(X_i - \mu_D)^\top$ be the empirical second moment with the actual mean of the distribution, and again define $\widehat{M}_W$ analogously. By Theorem A.1, by our choice of $n$, we have

$$ \|\widehat{M}_D - \Sigma_D\|_2 \leq \frac{1}{4}\|\Sigma_D\|_2 , \text{ and} $$
$$ \|\widehat{M}_W - \Sigma_W\|_2 \leq \frac{1}{4}\|\Sigma_W\|_2 , $$

both with probability at least $99/100$. Thus, by a union bound, all these events happen simultaneously with probability at least $9/10$. Condition on the event that all these events occur. Then this implies that

$$ \|\widehat{M}_D\|_2 \leq \frac{5}{4}\sigma^2 , \text{ and } \|\widehat{M}_W\|_2 \leq \frac{5}{4}\sigma^2 . $$

Since $\widehat{\Sigma}_D \preceq \widehat{M}_D$ and $\widehat{\Sigma}_W \preceq \widehat{M}_W$, we conclude that

$$ \|\widehat{\Sigma}_D\|_2 \leq \frac{5}{4}\sigma^2 , \text{ and } \|\widehat{\Sigma}_W\|_2 \leq \frac{5}{4}\sigma^2 . $$

The result then follows by applying Lemma 3.1 to the empirical distributions over $\mathcal{X}$ and $\mathcal{W}$, respectively. $\quad\square$

# B  Additional Experiments

In this section, we provide various additional experiments. First, in Table 4, we provide a wider variety of attack parameters for our main experimental setup. Then, in Table 5, we present our results training a VGG model instead of a Resnet model.

Table 4: Full table of accuracy and number of poisoned images left for different attack parameters. For each attack to target label pair, we provide a few experimental runs with different backdoor.

| Sample | Target | Epsilon | Nat 1 | Pois 1 | # Pois Left | Nat 2 | Pois 2 | Std Pois |
|---|---|---|---|---|---|---|---|---|
|  | bird | 5% | 92.27% | 74.20% | 57 | 92.64% | 2.00% | 1.20% |
| | | 10% | 92.32% | 89.80% | 7 | 92.68% | 1.50% | |
|  | bird | 5% | 92.49% | 98.50% | 0 | 92.76% | 2.00% | 1.90% |
| | | 10% | 92.55% | 99.10% | 0 | 92.89% | 0.60% | |
|  | bird | 5% | 92.66% | 89.50% | 14 | 92.59% | 1.40% | 1.10% |
| | | 10% | 92.63% | 95.50% | 2 | 92.77% | 0.90% | |
|  | cat | 5% | 92.45% | 83.30% | 24 | 92.24% | 0.20% | 0.10% |
| | | 10% | 92.39% | 92.00% | 0 | 92.44% | 0.00% | |
|  | cat | 5% | 92.60% | 95.10% | 1 | 92.51% | 0.10% | 0.10% |
| | | 10% | 92.83% | 97.70% | 1 | 92.42% | 0.00% | |
|  | cat | 5% | 92.80% | 96.50% | 0 | 92.77% | 0.10% | 0.00% |
| | | 10% | 92.74% | 99.70% | 0 | 92.71% | 0.00% | |
|  | dog | 5% | 92.91% | 98.70% | 0 | 92.59% | 0.00% | 0.00% |
| | | 10% | 92.51% | 99.30% | 0 | 92.66% | 0.10% | |
|  | dog | 5% | 92.17% | 89.80% | 7 | 93.01% | 0.00% | 0.00% |
| | | 10% | 92.55% | 94.30% | 1 | 92.64% | 0.00% | |
|  | horse | 5% | 92.38% | 96.60% | 0 | 92.87% | 0.80% | 0.80% |
| | | 10% | 92.72% | 99.40% | 0 | 93.02% | 0.40% | |
|  | horse | 5% | 92.60% | 99.80% | 0 | 92.57% | 1.00% | 0.80% |
| | | 10% | 92.26% | 99.80% | 0 | 92.63% | 1.20% | |
|  | cat | 5% | 92.68% | 97.60% | 1 | 92.72% | 8.20% | 7.20% |
| | | 10% | 92.59% | 99.00% | 4 | 92.80% | 7.10% | |
|  | cat | 5% | 92.86% | 98.60% | 0 | 92.79% | 8.30% | 8.00% |
| | | 10% | 92.29% | 99.10% | 0 | 92.57% | 8.20% | |
|  | deer | 5% | 92.68% | 99.30% | 0 | 92.68% | 1.10% | 1.00% |
| | | 10% | 92.68% | 99.90% | 0 | 92.74% | 1.60% | |
|  | deer | 5% | 93.25% | 97.00% | 1 | 92.75% | 2.60% | 1.10% |
| | | 10% | 92.31% | 97.60% | 1 | 93.03% | 1.60% | |
|  | frog | 5% | 92.87% | 88.80% | 10 | 92.61% | 0.10% | 0.30% |
| | | 10% | 92.82% | 93.70% | 3 | 92.74% | 0.10% | |
|  | frog | 5% | 92.79% | 99.60% | 0 | 92.71% | 0.20% | 0.20% |
| | | 10% | 92.49% | 99.90% | 0 | 92.58% | 0.00% | |
|  | bird | 5% | 92.52% | 97.90% | 0 | 92.69% | 0.00% | 0.00% |
| | | 10% | 92.68% | 99.30% | 0 | 92.45% | 0.50% | |
|  | bird | 5% | 92.51% | 87.80% | 1 | 92.66% | 0.20% | 0.00% |
| | | 10% | 92.74% | 94.40% | 0 | 92.91% | 0.10% | |

Table 5: Full table of accuracy and number of poisoned images left for different attack parameters for a VGG model. For each attack to target label pair, we provide a few experimental runs with different backdoors. Here, we present results for $\varepsilon = 10\%$ because unlike our results for Resnet, in many cases 5% poisoned images was not enough to install the backdoor.

| Sample | Target | Epsilon | Nat 1 | Pois 1 | # Pois Left | Nat 2 | Pois 2 | Std Pois |
|---|---|---|---|---|---|---|---|---|
|  | bird | 10% | 92.82% | 83.50% | 23 | 92.64% | 2.00% | 1.30% |
|  | bird | 10% | 93.73% | 98.40% | 0 | 93.17% | 1.00% | 1.40% |
|  | bird | 10% | 93.30% | 96.60% | 1 | 93.63% | 0.90% | 0.80% |
|  | cat | 10% | 93.39% | 82.90% | 12 | 92.24% | 0.20% | 0.30% |
|  | cat | 10% | 93.16% | 99.10% | 0 | 93.43% | 0.10% | 0.20% |
|  | cat | 10% | 92.90% | 99.40% | 0 | 93.17% | 0.00% | 0.60% |
|  | dog | 10% | 93.21% | 99.90% | 0 | 93.35% | 0.00% | 0.10% |
|  | dog | 10% | 93.20% | 92.20% | 2 | 93.32% | 0.10% | 0.00% |
|  | horse | 10% | 93.12% | 99.60% | 0 | 93.28% | 0.40% | 0.50% |
|  | horse | 10% | 92.95% | 99.90% | 0 | 93.13% | 1.00% | 0.80% |
|  | cat | 10% | 93.15% | 97.20% | 0 | 93.12% | 7.20% | 7.60% |
|  | cat | 10% | 93.15% | 99.80% | 0 | 93.27% | 6.90% | 8.60% |
|  | deer | 10% | 93.18% | 99.30% | 0 | 93.10% | 1.80% | 1.40% |
|  | deer | 10% | 93.26% | 99.20% | 0 | 93.04% | 1.50% | 1.20% |
|  | frog | 10% | 93.33% | 89.80% | 9 | 93.51% | 0.30% | 0.00% |
|  | frog | 10% | 92.90% | 99.80% | 0 | 93.24% | 0.10% | 0.00% |
|  | bird | 10% | 93.47% | 98.40% | 0 | 93.44% | 0.00% | 0.10% |
|  | bird | 10% | 93.20% | 93.40% | 0 | 92.95% | 0.00% | 0.10% |