[Reviews · NeurIPS 2018]

Reviewer 1



This paper provides a methodology called spectral signatures that analyzes the covariance of the feature representation learnt by the neural network to detect backdoor attacks. Detecting backdoor attacks in neural networks is an important and very recent problem that hasn't been solved properly. The paper is well-written and easy to follow. However, I found that the evaluation is limited and there are some inconsistencies in the paper.  After reading the rebuttal, I am still concerned by the limitations in the experimental evaluation. Namely, the authors did not test to see whether the algorithm would be tricked by benign outliers or strongly multimodal distributions. While Great Danes are quite different from chihuahuas, there is somewhat of a spectrum of dog breeds between them that would not really make chihuahuas outliers in the data. In other words, we agree that the distribution has high variance but not that it is multimodal or contains subpopulations in that sense. Additionally, they did not consider how an adversary may try to circumvent the defense. All in all, we still feel that it is just below the threshold because the authors haven't really shown that the method is robust to different types of data and/or poisoning attempts.   • The proposed methodology assumes that a single unpoisoned class does not contain subpopulations (Def. 3.1) the methodology relies on the ability of identifying differences in the means of subpopulations in the same class (poison vs. legitimate). However,  classification tasks may include multiple subpopulations in the same class e.g., mammals may contain bears, dogs, cats. I suggest that the authors to address this particular concern and clarify when their methodology works. Ideally, some experimental results should be provided if they claim their solution works for the above example. • The authors contradict themselves in the experimental results reported:  o In line 112: "With as few as 250 (5% of a chosen label) poisoned examples, we successfully achieve both of the above goals on the CIFAR-10 dataset." o In line 258: "As an aside, we note that 5% poisoned images is not enough to capture the watermark according to our definition in our examples from Figure 2, but 10% is sufficient." • The paper calls the backdoor a "watermark". A backdoor is not necessarily a watermark. For example, in the original paper by Gu et al. (2017) that introduced backdoor attacks, the poisonous images of stop signs contained very visible stickers on them. Furthermore, a watermark has a different meaning in the security community and it threw me off a little while I was reading. I suggest changing watermark to backdoor or backdoor trigger throughout the paper.  • For readability purposes, it would be good to label columns 4th and 6th in table 2.  • Finally, using a single dataset and network architecture to test the approach is a bit limiting to fully generalize the results.  

Reviewer 2



In this paper, the authors have presented the notion of spectral signatures and demonstrate how they can be used to detect backdoor poisoning attacks. Our outlier detection method relies on the idea that learned representations for classifiers amplify signals crucial to classification. To me, the idea makes sense and have some promising results. I have only one concern for this paper. If SVD is employed for the method, what is the computational complexity of the proposed model? Can the method be applied to large-scale real-world applications?

Reviewer 3



The authors demonstrate how SVD on learned representations can be used to remove poisoned examples from the dataset. Perhaps the most interesting result is that the watermark is enhanced through training in higher layers of the network and that this is a function of how models are overparameterized. This leads me to wonder about the effects of distillation on these types of attacks. For example, if a trained model was distilled into a much smaller representation would this separation technique still hold? Phrased differently/relatedly, I assume these effects change as a function of (1 − β)m? It would be great to see an experiment in which the number of filters is varied and test how robust this approach is to those changes.

Reviewer 4



The paper presents a novel idea to detect backdoor attacks in neural networks. Although the idea is interesting, the experimental evaluation presented is limited in comparison to accepted papers in similar fields in top conferences. To ensure that the paper makes the threshold, additional experiments are needed. In particular, new experiments that 1) assess what potential adversaries may attempt to do to circumvent the defense, 2) evaluate multiple DNN architectures, 3) explore the effects of sub-populations within the same class. It seems that the defense relies in finding the difference of population means; it is not clear if having a classification task, e.g. mammals vs. objects, would render the solution unusable.